# Split Aptamers Immobilized on Polymer Brushes Integrated in a Lab-on-Chip System Based on an Array of Amorphous Silicon Photosensors: A Novel Sensor Assay

**DOI:** 10.3390/ma14237210

**Published:** 2021-11-26

**Authors:** Manasa Nandimandalam, Francesca Costantini, Nicola Lovecchio, Lorenzo Iannascoli, Augusto Nascetti, Giampiero de Cesare, Domenico Caputo, Cesare Manetti

**Affiliations:** 1Department of Environmental Biology, Sapienza University of Rome, P.le Aldo Moro 5, 00185 Rome, Italy; nandimandalam.1873424@studenti.uniroma1.it (M.N.); cesare.manetti@uniroma1.it (C.M.); 2CREA-DC Research Centre for Plant Protection and Certification, 00156 Rome, Italy; 3Department of Information, Electronic and Telecommunication Engineering, Sapienza University of Rome, Via Eudossiana 18, 00184 Roma, Italy; nicola.lovecchio@uniroma1.it (N.L.); lorenzo.iannascoli@uniroma1.it (L.I.); giampiero.decesare@uniroma1.it (G.d.C.); domenico.caputo@uniroma1.it (D.C.); 4School of Aerospace Engineering, Sapienza University of Rome, Via Salaria 851/881, 00138 Rome, Italy; augusto.nascetti@uniroma1.it

**Keywords:** split aptamers, polymer brushes, ATP, a-Si:H photosensors, Lab-on-Chip

## Abstract

Innovative materials for the integration of aptamers in Lab-on-Chip systems are important for the development of miniaturized portable devices in the field of health-care and diagnostics. Herein we highlight a general method to tailor an aptamer sequence in two subunits that are randomly immobilized into a layer of polymer brushes grown on the internal surface of microfluidic channels, optically aligned with an array of amorphous silicon photosensors for the detection of fluorescence. Our approach relies on the use of split aptamer sequences maintaining their binding affinity to the target molecule. After binding the target molecule, the fragments, separately immobilized to the brush layer, form an assembled structure that in presence of a “light switching” complex [Ru(phen)_2_(dppz)]^2+^, emit a fluorescent signal detected by the photosensors positioned underneath. The fluorescent intensity is proportional to the concentration of the target molecule. As proof of principle, we selected fragments derived from an aptamer sequence with binding affinity towards ATP. Using this assay, a limit of detection down to 0.9 µM ATP has been achieved. The sensitivity is compared with an assay where the original aptamer sequence is used. The possibility to re-use both the aptamer assays for several times is demonstrated.

## 1. Introduction

Aptamers are single standard DNA or RNA sequences that have been shown to bind non-nucleic acid target molecules with high affinity and specificity [1]. Due to the simplicity of synthesis and rapid production process, aptamers are considered as a valid alternative to antibodies or other bio-mimetic receptors for the development of biosensors [2].

Aptamers can be used in analytical chemistry either in solution or immobilized in heterogenous assay. When an aptamer is bound to a material which acts as a transducing element we can define it an aptasensor [2]. Recently, nanoscience and nanotechnology has generated novel nanomaterials which results in aptasensor systems with many applications in the field of health-care and diagnostics [3,4,5]. Optical detection methods such as colorimetry, fluorescence, surface-enhanced Raman scattering (SERS), and surface plasmon resonance (SPR) are widely applied for signal detection in aptasensors because of their ease of use and high sensitivity [6,7,8,9]. In particular, fluorescence detection is a versatile, high-sensitive technique in quantitative analysis and is specially suitable for detection of the conformational change occurring upon the interaction of the target with the aptasensor system [6]. Aptasensing based on fluorescent detection methods have been achieved by using either fluorescent labeled and label-free aptamers [10,11]. In this last case, there is no need of chemically modifying the aptamers with the fluorophore, which eliminates time-consuming, labor-intensive, and costly synthesis processes [12,13].

Nanomaterials used for chemical modification of microfluidic devices to obtain lab-on-chip systems have been reported [14]. The advantages of using these devices are the miniaturization, which allows to use lower assay volumes, less energy and time and portability. More importantly, due to their small size, microfluidic platforms improve the transport of the analyte molecules to the biorecognition elements improving the sensor response time and in some cases the sensitivity [15]. Aptamers have been immobilized into microfluidic devices using different nanomaterials [16] such as nanoparticles [17], graphene oxide [18], self-assembled monolayers and polymer brushes for the detections of different targets such as small molecules and biomarkers [14,19,20,21,22,23], pathogens [24], and cancer cells [25,26,27]. The nanomaterials applied in the above systems act as transducing agents and the signal deriving from the interaction of the target and the immobilized aptamers is detected. Detection occurs often on-chip when electrochemistry is used, while when optical methods are applied, in most cases, a fluorescent microscope is coupled to the device [14].

In previous publications, we reported the development of functionalized microfluidic channels for the detection of Ochratoxin A and adenosine triphosphate (ATP) using a Lab-on-chip (LoC) in which the detection occurs through an array of amorphous silicon photosensors (a-Si:H) optically aligned with the functionalized microfluidic network [20,21,22].

In this work we show the growth of a polymer brush layer in a microfluidic network that is functionalized with aptamer fragments [28], as a novel method for the detection of ATP in LoC systems. ATP is an universal marker for detecting the presence of pathogens [29] and here is used as proof of principle to test the assay based on aptamer fragments. The novelty of this aptasensor material is based on the random immobilization of the aptamer fragments, having affinity toward ATP, into the polymer brush layer of 2-hydroxyethyl methacrylate functional units. Upon interaction with ATP these form an assembled structure emitting fluorescence when the fluorophore [Ru(phen)_2_(dppz)]^2+^ is present (Figure 1). The fluorescence signal is detected by the array of a-Si:H photosensors coupled with the functionalized microfluidic system. The random immobilization of the fragments along the multifunctional units of the PHEMA layer gives rise to the recognition element, which is dynamic, generating a fluorescence signal only upon interaction with ATP. The performance of the this assay is compared with that based on the ATP original aptamer sequence [30] immobilized into the brush functionalized channels and with others reported LoCs. The possible regeneration of the aptasensing brush layer for several measurements is exhibited for both fragments and original aptamer assays. Finally, since the detection is integrated with the assay, the portability of this LoC is also demonstrated, implying wide applicability for biosensing.

## 2. Materials and Methods

### 2.1. Reagents and Equipment

All reagents were purchased from Merck (Darmstadt, Germany). Pressure sensitive adhesives (ARCare, ARClad IS-7876 and ARFlow) were purchased from Adhesive Research (Limerick, Ireland), and are cut with a desktop cutting plotter (Silhouette Curio, Graphtec, Italy).

2-Hydroxyethtil methacrylate (HEMA) was distilled prior to use, whereas the other chemicals were used without further purification. 2-Bromo-2-methyl-propionic acid 3-trichlorosylanyl-propyl ester (BMPTS) is synthesized following a previously reported procedure [31]. Acetone and ethanol 96% (analytical reagent grade) were used without further purification, while toluene was distilled over sodium. Water was purified with a Milli-Q pulse (MILLIPORE, R = 18.2 MΩ·cm) ultra-pure water system.

Tris buffer: Tris 12.5 mM, NaCl 150 mM and KCl 25 mM, pH 7.2; Tris-HCl buffer: Tris 10 mM and MgCl_2_ 220 mM, pH 7.0; PBS buffer: NaCl 1.37 mM, KCl 2.7 mM, Na_2_HPO_4_ 10 mM, KH_2_PO_4_ 1.76 mM, pH 7.5.

Aptamers were purchased by Merck:


**Aptamers**

**Sequence**
ATP-aptamer5′-H2N-TTTTTACCTGGGGGAGTATTGCGGAGGAAGG-3′R1-aptamer5′-H2N-TTTTTACCTGGGGGAGT-3′P1-apatmer5′-H2N-GACTACGGTGATTTTTGCGGAGGAAGGT-3′

Rutenium 1,10-phenatroline dipirido phenazine ([Ru(phen)_2_(dppz)]^2+^) was synthesized according to the reported procedure [30].

Aptamers were dissolved in PBS buffer and a solution of 0.3 mg/mL was prepared. Prior to immobilization they were kept for 1 min at 95 °C and left to cool down at room temperature.

### 2.2. Fabrication and Functionalization of the Microfluidic Chip with ATP Amino-Aptamer and R1/P1-Fragments

Bare glass slides with dimensions 4 × 5 cm^2^ were treated with piranha solution (H_2_SO_4_:H_2_O, 3:1) for 30 min, then rinsed with Milli Q water and dried under a stream of nitrogen. The clean glass slides are then incubated in a solution of 0.01% of 3-trichlorosylanyl-propyl ester (BMPTS) in dry toluene overnight [31]. The formed self-assembled monolayer was rinsed with dry toluene, acetone, and ethanol and finally dried with a stream of nitrogen. To develop a PHEMA polymer film, a solution of 15 mL of HEMA and 15 mL of Milli Q water was bubbled with nitrogen gas for 30 min and this solution is introduced into a mixture of copper(I) chloride (0.082 g), copper(II) bromide (0.054 g), and 2,2′-dipyridyl (0.366 g) which was under argon gas. The above mixture with solution is stirred for about 10 min under argon gas until the particles are completely dissolved resulting in a dark brown solution. The solution is then transferred using a syringe on to the self-assembled monolayer under an argon environment and sealed for 4 h. The polymerized glass slides were removed and washed under ethanol and milli Q water then dried with a stream of nitrogen. The PHEMA polymer glass slides are then soaked in a solution of succinic anhydride (1.5 g) and triethylamine (1.5 mL) in a tetrahydrofuran (27 mL). The PHEMA polymer glass slides are functionalized with succinic anhydride (PHEMA-SA) to obtain carboxylic moieties (PHEMA-SA). After 18 h the samples are rinsed with ethanol and dried with a stream of nitrogen. After this step, a PSA layer (1) (Figure 1) which consists of an array of ten channels is attached on the glass slide completely coated with PHEMA functionalized with succinic anhydride (PHEMA-SA). The PSA layer is made of AR-CARE that is cut by using the plotter in the array of ten channels with the following dimensions: 1.6 mm width and 20 mm length while the height corresponds to that of the PSA, which is 150 µm (Figure 1).

A water solution (2 mL) containing 15 mg of n-hydroxysuccinimide (NHS) and 75 mg of 1-ethyl-3-(3-dimethylaminopropyl) carbodiimide (EDC)) is spotted on the channels and left to react for 1 h to obtain the NH-ester functional groups (PHEMA-NHS). The samples are then rinsed with water and dried with a stream of nitrogen. Finally, a solution of PBS containing ATP amino-aptamer (0.3 mg/mL) or alternately a mix of R1/P1-apatamer fragments (0.3 mg/mL) is spotted on the channels of PHEMA-NHS and incubated overnight at 4 °C. The obtained polymer films PHEMA-aptamer or PHEMA-aptamer fragments are rinsed with PBS. Finally, the substrate is dried with a stream of nitrogen. Afterwards, the unreacted NHS-groups are blocked by spotting on the samples a solution of ethanolamine 10 mM in PBS pH 7, for 30 min and subsequently PHEMA-aptamer or PHEMA-aptamer fragments are rinsed with Tris and Tris-HCl buffer, respectively. Finally, the channels are rinsed with tris buffer and dried with a stream of nitrogen and the device is assembled (Figure 1): the intermediate layer AR-FLOW (2) with via-holes between bottom and the top layers, the top layer (AR-CARE) defining the inlet distribution channel (3), the chip cover (AR-FLOW) with access hole (4) for the inlet (the largest hole) and the outlets (the ten small holes on both sides) are sealed to the functionalized substrates (Figure 1) to obtain the final microfluidic chip array.

### 2.3. Structure of Lab-on-Chip System

The lab-on-chip consists of a metallic box (Figure 2a) containing on the top side of the lid an access hole for inserting a Gilson pipette and in the internal part a light emitting diode (LED) (Figure 2b). In the metallic box is contained the electronic board (Figure 2c) to which is connected the array of a-Si:H photosensors (Figure 2d). The functionalized microfluidic network (Figure 2e) is then coupled with the array of a-Si:H photosensors. The a-Si:H photosensors consist of an optoelectronic-on-glass substrate on a 50 × 50 × 1.1 mm^3^ Borofloat glass purchased from ZAOT s.r.l. (Milan, Italy), containing on one side an array of amorphous silicon (a-Si:H) photosensors and on the other side an interferential filter. The array is composed of 5 columns and 6 rows of photosensors. The filter rejects the excitation light and transmits almost unchanged the fluorescent radiation. In this way photosensors will detect only the re-emitted light. The photosensor fabrication and interferential filter deposition and performance were reported in previous publications [32,33].

The functionalized chip is optically coupled to the optoelectronic glass placing it above the interferential filter glass side (Figure 2d) and aligning the channels with the photosensor array columns. The fluorescence is induced by the LED (APG2C1-450 from Roithner Lasertechnik, Wien, Austria) positioned at 2.5 cm above of the microfluidic chip. A band-pass filter centered at 450 nm and bandwidth of ±10 nm (Corion Corp., Alliston, MA, USA) is placed between the LED and the microfluidic chip in order to remove the undesired wavelengths coming from the LED (Figure 2b). The value of the decreased or increased fluorescent signal was detected by the photosensors and read-out by a very low-noise electronic board [34]. The connection between the photosensor pads and the electronic board was achieved through a card edge connector [Samtec MB1-150-S-02-SL].

### 2.4. ATP Detection with Array of Channels Functionalized with PHEMA-Aptamer Fragments

The functionalized microfluidic network is optically aligned with the array of photosensors. By using a Gilson pipette, all the channels were filled with Tris-HCl buffer (120 µL) from the access hole, subsequently the photocurrent (F_0_) for the sensors positioned underneath each channel was measured. The device was then dried positioning a filter paper in correspondence of the outlets. Subsequently, the channels were filled with different solutions containing ATP (1–1000 µM) and [Ru(phen)_2_(dppz)]^2+^ alternatively 0.5–2 µM for different incubation time (10–20 min), inserting the solutions from the outlets. Subsequently, all the channels were rinsed about three times inserting Tris-HCl buffer (120 µL) in the access hole by using the Gilson pipette and driving the liquid out from the outlets utilizing the filter paper. Afterwards, the channels were filled with the Tris-HCl buffer from the access hole and the photocurrent (F_1_) was measured. The net fluorescent signal (F) measured resulting after the incubation of ATP with the aptasensor is F = F_1_ − F_0_.

The blank was taken inserting a solution containing ATP 1000 µM and [Ru(phen)_2_(dppz)]^2+^ alternatively 0.5–2 µM for different incubation time (10–20 min) in a channel where all the NHS-ester groups were blocked with ethanolamine 4 °C overnight. The limit of detection (LOD) was calculated as 3.3 σ/S, where σ is the standard deviation of the blank and S is the slope of the calibration curve. The standard deviation (σ) was taken over hundreds points of the blank signals.

### 2.5. ATP Detection with Array of Channels Functionalized with PHEMA-Aptamer

By using a Gilson pipette, all the channels were filled with a solution of [Ru(phen)_2_(dppz)]^2+^ in Tris buffer with a concentration varying between 0.5 and 2 µM, from the access hole, and incubated for 10 min. Subsequently, the channel were rinsed about ten times inserting Tris buffer (120 µL) from the access hole by using the Gilson pipette and driving the liquid out from the outlets with the filter paper. Afterwards, the channels were filled with the Tris buffer by the access hole and the photocurrent (F_0_) for the sensors positioned underneath each channel was measured. The device was then dried with a stream of nitrogen and the channels were filled from the outlets with different solutions of ATP (1–1000 µM) for different incubation time (10–20 min). Subsequently, by pipetting in the inlet hole 120 µL of Tris buffer (three times) the channels were rinsed again. The channels were filled again with the buffer by the access hole and the photocurrent (F_1_) was measured. The fluorescent signal (F) measured resulting after the interaction of ATP with the aptasensor is F = F_0_ − F_1_.

The blank was taken inserting a solution containing [Ru(phen)_2_(dppz)]^2+^ 0.5–2 µM, 10 min incubation time, and subsequently ATP 1000 µM for different incubation time (10–20 min) in a channel where all the NHS-ester groups were blocked with ethanolamine 4 °C overnight.

The limit of detection (LOD) was calculated as 3.3 σ/S, where σ is the standard deviation of the blank and S is the slope of the calibration curve. The standard deviation (σ) was taken over hundreds points of the blank signals.

## 3. Results and Discussion

In order to develop the novel polymer-brush based assay, split aptamers named fragment R1 and P1, for binding ATP were selected. The R1 e P1 fragments were randomly immobilized into a layer of PHEMA polymer brushes grown on a surface of a microfluidic network [21] forming PHEMA-aptamer fragments. Our hypothesis is that the fragments after binding ATP would assemble into a folded structure that in presence of [Ru(phen)_2_(dppz)]^2+^ would give a fluorescent signal proportional to the concentration of ATP present in solution. [Ru(phen)_2_(dppz)]^2+^ is a molecular light switching complex, which has no luminescence in aqueous solution but strong luminescence when intercalating into the non-aqueous pocket of DNA duplex [35]. A schematic representation of synthesis of the sensing system and the hypothesized mechanism of the fragments interaction after binding ATP is shown in Figure 1.

The performance of the split aptamer assay was compared with that of an original ATP-aptamer, also integrated in the microfluidic channel. The ATP-aptamer assay works with a different mechanism: the fluorophore [Ru(phen)_2_(dppz)]^2+^ is intercalated between the base pairs of the aptamer. Upon the interaction of ATP with the aptasensor, a change of the aptamer conformation causes the release of the fluorophore, yielding a decrease of fluorescent signal [30].

### ATP Detection with the Array of Channels Functionalized with PHEMA-Aptamer Fragments and PHEMA-Aptamer

As schematically reported in Figure 2, the LoC presented here consists of a metallic box containing a LED (Figure 2b) and the functionalized microfluidic network coupled with the array of a-Si:H photosensors, which is connected to an electronic board (Figure 2c). The channels functionalized with PHEMA-aptamer fragments are optically aligned with the array of a-Si:H photosensors. The system also includes a home-made apparatus which permits spotting a drop of 120 µL of solution directly into the access hole (Figure 2a,c) from where the solution moves into the array of channels by capillary force.

Upon the injection of the solution containing ATP and [Ru(phen)_2_(dppz)]^2+^, the immobilized split aptamers would form the assembled structure (Figure 1). The [Ru(phen)_2_(dppz)]^2+^ intercalated in the complex emits a fluorescent signal proportional to the concentration of ATP and is detected by the a-Si:H sensors as an increase of photocurrent.

In order to demonstrate our hypothesis, each functionalized channel is filled with solutions having different concentrations of ATP. Due to the redundancy of channels, the system allows having a duplicate measurement for each concentration analyzed. In order to optimize the assay, the concentration of [Ru(phen)_2_(dppz)]^2+^ was varied. It was found that [Ru(phen)_2_(dppz)]^2+^ over 3 µM concentration showed a decrease of the photocurrent, which indicates the light absorption by the fluorophore overcomes the fluorescence emission. This behavior is probably due to the not-specific absorption of the fluorophore into the brush layer, as observed also for the blank sample (channel where brush layer is functionalized with ethanolamine instead of aptamers). On the other hand, when a concentration of [Ru(phen)_2_(dppz)]^2+^ lower than 0.5 µM was used to perform the ATP analysis, an increase of photocurrent was observed only for solutions having ATP concentration between 100–1000 µM. Based on these observations, the assay was tested varying the concentration of [Ru(phen)_2_(dppz)]^2+^ between 0.5 and 2 µM. Moreover, the time was varied and we found that experiments with incubation lower than 10 min did not show reproducible results, when both the concentrations of ATP and [Ru(phen)_2_(dppz)]^2+^ were varied.

Initially, for testing the LoC functionalized with the PHEMA-aptamer fragments, solutions with different concentrations of ATP and [Ru(phen)_2_(dppz)]^2+^ were inserted in the channels from the outlets, and kept inside for 10 or 20 min. Subsequently, the metallic box was closed and 120 µL of Tris buffer were inserted for three times through the access hole (Figure 2a,c) for a complete rinsing. The solution is moved within the channel array by the capillary force driven by the filter paper positioned on the outlet holes. In this manner the excess of [Ru(phen)_2_(dppz)]^2+^, which was not included in the folded structure formed after reaction of the fragments with ATP, is rinsed away to avoid any interference with the measurement. An increase of the photocurrent was observed only when both ATP and [Ru(phen)_2_(dppz)]^2+^ were incubated for both 10 and 20 min in the channels functionalized with the fragments. On the other hand, when both ATP and [Ru(phen)_2_(dppz)]^2+^ were incubated in the channel functionalized only with ethanolamine instead of the fragments, the increase of photocurrent was not observed for the range of 10–20 min of incubation time. Moreover, when only ATP was inserted in channels functionalized with PHEMA-aptamer fragments, the photocurrent did not vary. Figure 3 reports the variation of photocurrent as a function of the ATP concentration both for the variation of [Ru(phen)_2_(dppz)]^2+^ concentration and incubation time. It is worth noticing that F_0_ is equal to (100 ± 0.32) pA, while the blank signal is equal to (99 ± 0.32) pA. The decrease of the blank signal is due to the [Ru(phen)_2_(dppz)]^2+^ absorption.

The data showed that for the concentration of [Ru(phen)_2_(dppz)]^2+^ of 1 and 2 µM, ATP was detected in the range 0.1–1000 µM, whereas when the concentration of [Ru(phen)_2_(dppz)]^2+^ 0.5 µM was used, ATP under 1 µM was not detected. The data were fitted with a straight line equation showing an R^2^ of 0.99 for all the curves. In all the experiments performed, the increase of the incubation time from 10 to 20 min did not improve the detection limit (LoD). According to the slopes of the curves and the standard deviation of the blank samples, the calculated LoDs were very similar ranging between 0.89 and 0.98 µM for all the experimental conditions, with the exception of the curve obtained using 0.5 µM [Ru(phen)_2_(dppz)]^2+^ for 20 min incubation time, which was calculated to be 1.5 µM. On the other hand, the slopes of the curves, obtained using [Ru(phen)_2_(dppz)]^2+^ 1 µM after 20 min incubation time, showed the best normalized sensitivity (S) of 1.19. This result suggests that, upon the formation of the assembled structure, the 0.5 µM [Ru(phen)_2_(dppz)]^2+^ concentration is not sufficient to achieve the maximum fluorescent signal. On the other hand, the sensitivity of the PHEMA-aptamer fragments was lower for [Ru(phen)_2_(dppz)]^2+^ 2 µM concentration compared to that observed for 1 µM. This behavior may be due to a partial not-specific absorption of the [Ru(phen)_2_(dppz)]^2+^ on the PHEMA, as that was detected for concentrations of [Ru(phen)_2_(dppz)]^2+^ higher than 3 µM.

In this work, we also compared the performance of the microfluidic system functionalized with the PHEMA-aptamer fragments and that with the PHEMA-aptamer. For performing ATP detection, the same experimental conditions were applied. In this assay, the channels were first incubated with solution of [Ru(phen)_2_(dppz)]^2+^ for 10 min, subsequently the channels were rinsed, by pipetting 120 µL of Tris buffer in the inlet hole, for ten times. The channels were then aligned with the array a-Si:H photosensors and the photocurrent was measured (F_0_). The channels were filled from the outlets with solutions having different concentrations of ATP and after rinsing with 120 µL of buffer for three times, the photocurrent was measured (F_1_). Upon ATP interaction with the immobilized PHEMA-aptamer, the change of conformation causes the release of the previously intercalated [Ru(phen)_2_(dppz)]^2+^ to bind ATP. As a consequence, a decrease of the photocurrent was observed. No photocurrent variation was detected in the channel functionalized only with ethanolamine. Moreover, when only ATP was inserted in channels functionalized with PHEMA-aptamer the photocurrent did not vary. Furthermore, for this assay, the concentration of [Ru(phen)_2_(dppz)]^2+^ was varied between 0.5 and 2 µM and the ATP incubation time between 10 and 20 min (Figure 4). In this case we found that F_0_ is equal to (107 ± 0.45) pA, while the blank signal is equal to (99 ± 0.45) pA.

The data were fitted with a straight line equation showing an R^2^ of 0.99 for all the curves. In all the experiments performed, the increase of the incubation time from 10 to 20 min did not improve the detection limit (LoD). According to the slopes of the curves and the standard deviation of the blank samples, the calculated LoDs yield, for all the experimental conditions, a very similar values ranging between 1.14 and 1.29 µM. The lowest LoD of 1.14 µM was found for 2 µM [Ru(phen)_2_(dppz)]^2+^ 20 min incubation time. The LoDs of this assay are slightly higher than those obtained using the chip functionalized with the ATP-fragment assay. Nevertheless, also considerations on the easiness of the assay has to be taken: the PHEMA-aptamer fragments assay results advantageous in terms of number of steps necessary to perform the analysis. In the PHEMA-aptamer system, the [Ru(phen)_2_(dppz)]^2+^ is first reacted with aptamer and only subsequently ATP can be detected, while in the PHEMA-aptamer fragment assay, ATP is measured in a single step. The LoDs found for the PHEMA- aptamer fragments and PHEMA-aptamer assays for the detection of ATP are comparable with other systems based on aptamers and optical detection methods (Table 1).

The specificity of the proposed assays was also evaluated. Specifically, the ATP analogs GTP, CTP, and UTP at same concentration of 100 μM are found to promote minimal photocurrent variation for both PHEMA-aptamer fragments and PHEMA-aptamer functionalized channels (Figure 5). In particular, the selected nucleotides display between 4–9% and 10–16% of the photocurrent detected for ATP, when analyzed using the assays PHEMA-aptamer fragments and PHEMA-aptamer, respectively. Indeed, the PHEMA-aptamer fragments assay showed a better specificity, especially toward GTP and CTP.

The possibility of re-using the functionalized channels and thus the regeneration of the assay were also investigated. Since the binding of ATP with the aptamers is not-covalent, change of pH, temperature, or salt concentration may interfere with the binding process. Based on that, a solution of 2 M NaCl, after the first ATP measurement, was inserted in the channel and incubated for 10 min. After a few rinsing steps, the solutions of ATP were incubated again by varying the concentration and incubation time, following the rinsing steps previously described. Figure 6 demonstrates that ATP can be detected at least after three regeneration steps. The third step of regeneration was performed after two months the channel functionalization was made, keeping the channels in the fridge at 4 °C. This experiment demonstrates the possible re-usability of the assay and the stability provided by the brush layer after fabrication. The values of the slopes for the calibration curve obtained with both the regenerated and the fresh-functionalized devices were equal within an experimental error of about 10%.

## 4. Conclusions

In this work, a novel aptasensor based on split aptamers by using a PHEMA brush layer grown in microfluidic channel was developed. This system relies on the integration of the functionalized microfluidic channels with an array of a-Si:H photosensors leading to a LoC system for ATP detection. The brush-layer material demonstrates to be a versatile system for the immobilization of aptamers permitting to employ the split aptamers to have a novel aptasensing system. The performance of this novel assay was compared with that of an original ATP-aptamer also immobilized in the channel through the brush layer. The ATP detection limit was similar in the two assays, nevertheless the PHEMA-aptamer fragments assay showed better performance in terms of selectivity and time necessary for the measurement execution, which are crucial for portable devices. The variation of [Ru(phen)_2_(dppz)]^2+^ concentration and ATP incubation time for both assay systems were investigated and the LoDs obtained showed the assay sensitivities are comparable with other aptasensing systems based on optical detection. The re-usability of the assays was demonstrated for both PHEMA-aptamer fragments and PHEMA-aptamer assays. The LoC system presented here may have wide range of applicability as a portable sensing system in the field of health-care and diagnostics. In order to achieve a stand-alone system, further developments are focused on the integration of an extraction module directly connected with the presented LoC.

## Data Availability

Not Applicable.

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
