# Peer review of "Split Aptamers Immobilized on Polymer Brushes Integrated in a Lab-on-Chip System Based on an Array of Amorphous Silicon Photosensors: A Novel Sensor Assay"

_materials, 2021, doi:10.3390/ma14237210_

Round 1

Reviewer 1 Report

Summary:
This manuscript present a novel sensor assay with splitting aptamer sequence in lab-on-chip systems. There are a few places of the manuscript could be further polished, such as the structure/organization of the introduction. 

Major Review:

1. The structure of the 3rd paragraph of introduction seems a little bit unbalanced. Maybe the authors can split the 3rd paragraph of introduction into two parts: the description of the research field and the presented work. 

2. The authors may draw a chart to show the usage of the lab-on-chip system for section 2.3 (e.g. an overall flow-chart to demonstrate more details of each step for the reported work).

3. For figure 3, can the authors explain a little bit more detail why 20 min incubation has slightly higher F value for 1 uM, which is different with 0.5 uM and 2 uM?

Minor Review:
1. Page 1: Abstract
After binding the target molecule, the fragments, separately immobilized to the brush layer, form a assembled structure ...
a assembled  => an assembled 

2. Fig2:
It's better that the author draw another figure with more detailed view (e.g. top-view or side-view) as a sub-figure to show more detailed mechanism of the system.

3. Page 7 & 8: Figure 3 & 4
The legends are a little bit small and blur.

4. Page 9: Table 1
The line numbers have been messed up with the table contents

Author Response

Reviewer n. 1:

Summary:
This manuscript present a novel sensor assay with splitting aptamer sequence in lab-on-chip systems. There are a few places of the manuscript could be further polished, such as the structure/organization of the introduction. 

Major Review:

  1. The structure of the 3rd paragraph of introduction seems a little bit unbalanced. Maybe the authors can split the 3rd paragraph of introduction into two parts: the description of the research field and the presented work. 

Thanks for the comment. The third paragraph was changed to balance the information about the research fields and the present work.

  1. The authors may draw a chart to show the usage of the lab-on-chip system for section 2.3 (e.g. an overall flow-chart to demonstrate more details of each step for the reported work).

Thanks for the comment. A new Figure 2 was inserted to show how the LoC system works.

  1. For figure 3, can the authors explain a little bit more detail why 20 min incubation has slightly higher F value for 1 uM, which is different with 0.5 uM and 2 uM?

Thanks for the comment. To answer the question we inserted the following sentence in the “Results and Discussion”: “….This result suggests that, upon the formation of the assemble structure, the 0.5 µM [Ru(phen)2(dppz)]2+concentration is not sufficient to achieve the maximum fluorescent signal. On the other hand, the sensitivity of the PHEMA-aptamer fragments was lower for [Ru(phen)2(dppz)]2+ 2 µM concentration compared to that observed for 1 µM. This behavior may be due to a partial not-specific absorption of the [Ru(phen)2(dppz)]2+ on the PHEMA, as that was detected for concentrations of [Ru(phen)2(dppz)]2+ higher than 3 µM.”

Minor Review:

  1. Page 1: Abstract

After binding the target molecule, the fragments, separately immobilized to the brush layer, form a assembled structure ...
a assembled  => an assembled 

Thanks for the comment. The correction was done.

  1. Fig2:
    It's better that the author draw another figure with more detailed view (e.g. top-view or side-view) as a sub-figure to show more detailed mechanism of the system.

Thanks for the comment. Figure 2 became Figure 1 to address the comment of another reviewer and we added the cross section to better show the microfluidic system. Moreover, we inserted a Figure 2 to add more details of the LoC device.

  1. Page 7 & 8: Figure 3 & 4
    The legends are a little bit small and blur.

Thanks for the comment. Legend was corrected.

  1. Page 9: Table 1

The line numbers have been messed up with the table contents
Thanks for the comment. The table was corrected.

Reviewer 2 Report

Initially, I would like to congratulate the authors for the submitted manuscript. My considerations after careful reading are presented below.

  1. The manuscript is well structured, and has scientific importance, as it has developed new material with high sensitivity.
  2. The manuscript is well structured, and has scientific importance, as it has developed new material with high sensitivity.
  3. Aptamers were purchased by Merck, insert the CAS.
  4. Suggestion: address in the manuscript the selectivity of the proposed device.
  5. Suggestion: address in the manuscript the selectivity of the proposed device. Mainly compounds with similar chemical structure.

I recommend publishing after minor revisions.

Author Response

Reviewer n.2

Initially, I would like to congratulate the authors for the submitted manuscript. My considerations after careful reading are presented below.

  1. The manuscript is well structured, and has scientific importance, as it has developed new material with high sensitivity.

All authors thank the reviewer for the positive comments about the work reported in this manuscript.

  1. Aptamers were purchased by Merck, insert the CAS.

The aptamers used in this work were custom-made by Merck following the sequence reported in the “Material and Method section”, thus a CAS number is not reported by the company.

  1. Suggestion: address in the manuscript the selectivity of the proposed device.

Thanks for the suggestion. The selectivity, by definition is: “..the extent to which it can determine particular analyte(s) in a complex mixture without interference from other components in the mixture” (Journal of Pharmaceutical and Biomedical Analysis 14 (1996) 867-869). ATP is a marker used for detecting presence of pathogens, so a complex mixture could be, for example,  a solution containing bacteria. Nevertheless, in this work we study the PHEMA as material for developing a novel aptamer assay for the detection of ATP as a proof of principle and we analyzed the specificity which is the ability to assess unequivocally the analyte in the presence of components which may be expected to be present in a possible real solution to be analyzed (Journal of Pharmaceutical and Biomedical Analysis 14 (1996) 867-869).  Experiment of specificity conducted here are necessary to address the possible use of the novel assay. Further work will be focus on the analysis of real sample with this LoC system and then on the selectivity. Following the right comment of the reviewer we changed the sentence in the manuscript in the “Results and Discussion section”: …..“ In particular, the selected nucleotides display between 4-9% and 10-16% of the photocurrent detected for ATP, whenanalyzed using the assays PHEMA-aptamer fragments and PHEMA-aptamer, respectively.  Thus the PHEMA-aptamer fragment system showed a better specificity, especially toward GTP and CTP”.

I recommend publishing after minor revisions.

Reviewer 3 Report

I read the manuscript written by the authors. It is definitely an interesting subject and authors have performed an intensive long work for this. I have some remarks which needs to be addressed before publication:

1) In introduction, what do the authors mean in line 46-50 (page 2)? The lines are not written clearly.

2) In line 57, authors should clearly mention the details of the nanomaterials which are referred in the text. Clear specifications are missing. For example, nanoparticles is a general term and microbeads are not nanomaterials. Self assembled monolayers and polymer brushes are referred without examples.

3) Authors mention that they have previously published work on similar topic (Ref 20-22). What are the advantages of using polymer brushes in comparison with previous studies?

4) A schematic for the mechanism discussed in introduction will be useful for the readers

5) Authors should add real pictures of their prepared devices and lab on a chip system with photo-detectors.

6) In Figure 3 and 4, I miss the corresponding data for the blank measurements. It is difficult to compare and see improvements due to the performed modifications in the LoC devices.

7) I assume that the figure 3 refers to the data of LoC system functionalized with R1/P1 aptamer. However, the caption indicates ATP-fragments. Do the authors mean R1/P1 aptamer fragments? The multiple nomenclature can be confusing.

8) What was the reason for choosing the incubation time of 10-20 min?

9) Was there a change in calibration curve in the regeneration studies?

10) Authors have concluded that both of the devices showed similar results. But also mentioned that the selectivity and necessary time was better in ATP-fragments/R1-P1 Fragment. Authors should quantify the differences in the devices.

11) What are the authors’ recommendation for further developments in these LoC devices?

12) A final remark, there are quite a few corrections required from English perspective. Authors should proof-check the manuscript again.

Author Response

Reviewer n.3

I read the manuscript written by the authors. It is definitely an interesting subject and authors have performed an intensive long work for this. I have some remarks which needs to be addressed before publication:

All authors thank the reviewer for the positive comments about the work reported in this manuscript.

Comments:

  1. In introduction, what do the authors mean in line 46-50 (page 2)? The lines are not written clearly.

Thanks for the comment. The lines have been corrected inserting the following sentence: ” In particular, fluorescence detection is a versatile, high-sensitive technique in quantitative analysis and is specially suitable for detection of the conformational change occurring upon the interaction of the target with the aptasensor system [6]. Aptasensing based on fluorescent detection methods have been achieved by using either fluorescent labeled and label-free aptamers [10,11]”.

  1. In line 57, authors should clearly mention the details of the nanomaterials which are referred in the text. Clear specifications are missing. For example, nanoparticles is a general term and microbeads are not nanomaterials. Self assembled monolayers and polymer brushes are referred without examples.

Thanks for the comment. Including microbeads was our mistake. They are indeed not included within the nanomaterial to which we want refer to. We adjusted the references, which clarify how polymer brush layers and self-assemble monolayer have been applied to functionalize microfluidic channels with aptamers.

  1. Authors mention that they have previously published work on similar topic (Ref 20-22). What are the advantages of using polymer brushes in comparison with previous studies?

Thanks for the comment. In the previous publications the brushes were used to functionalized microfluidic channel in order to immobilize a recognition elements already define, such as an aptamer or an antibody or an enzyme. The different here is that, the random immobilization of the fragments along the multifunctional units of the PHEMA layer gives rise to the recognition element, which is dynamic, generating a fluorescence signal only upon interaction with ATP. To better explain this we added the following sentence in the “Introduction”:“The random immobilization of the fragments along the multifunctional units of the PHEMA layer gives rise to the recognition element, which is dynamic, generating a fluorescence signal only upon interaction with ATP”.

  1. A schematic for the mechanism discussed in introduction will be useful for the readers

Thanks for the comments. In order to clarify the mechanism for the reader we moved scheme 1 in the “Introduction”.

  1. Authors should add real pictures of their prepared devices and lab on a chip system with photo-detectors.

Thanks for the suggestion. The picture of the real LoC has been inserted as Figure 2 in the manuscript.

  1. In Figure 3 and 4, I miss the corresponding data for the blank measurements. It is difficult to compare and see improvements due to the performed modifications in the LoC devices.

Thanks for the comment. The values were added in the “Results and Discussion”: “It is worth noticing that F0 is equal to (100 ± 0.32) pA, while the blank signal is equal to (99 ± 0.32) pA. The decrease of the blank signal is due to the [Ru(phen)2(dppz)]2+ absorption.” and “In this case we found that F0 is equal to 107 ± 0.45 pA, while the blank signal keeps equal to 99 ± 0.45 pA.”

  1. I assume that the figure 3 refers to the data of LoC system functionalized with R1/P1 aptamer. However, the caption indicates ATP-fragments. Do the authors mean R1/P1 aptamer fragments? The multiple nomenclature can be confusing.

Thanks for the comment. The nomenclature was corrected over all the manuscript.

  1. What was the reason for choosing the incubation time of 10-20 min?

Thanks for the comments. In order to answer the question of the reviewer we inserted the following sentence in the “Results and Discussion”:  “…Also the time was varied and we found that experiments with incubation lower than 10 min did not show reproducible results when both the concentrations of ATP and [Ru(phen)2(dppz)]2+ were varied”.

  1. Was there a change in calibration curve in the regeneration studies?

Thanks for the comments. We have checked the slope of the calibration curve of the regenerated devices and we did not found any significant change. We have added the following phrase in the “Results and discussion section: “The values of the slopes for the calibration curve obtained with both the regenerated and fresh-functionalized devices were equal within an experimental error of about 10%.”

  1. Authors have concluded that both of the devices showed similar results. But also mentioned that the selectivity and necessary time was better in ATP-fragments/R1-P1 Fragment. Authors should quantify the differences in the devices.

Thanks for the comment. The different specificity between the two assay was quantified and the following sentence was introduced in the “Results and Discussion” section: In particular, the selected nucleotides display between 4-9% and 10-16% of the photocurrent detected for ATP, when analyzed using the assays PHEMA-aptamer fragments and PHEMA-aptamer, respectively. Indeed, the PHEMA-aptamer fragment system showed a better specificity, especially toward GTP and CTP”.

  1. What are the authors’ recommendation for further developments in these LoC devices?

Thanks for the comment. To give an idea of the possible prospectives of the system, we added the following phrase to the section “Conclusion”: “In order to achieve a stand-alone system, further development are focused on the integration of an extraction module directly connected with the presented LoC.”

  1. A final remark, there are quite a few corrections required from English perspective. Authors should proof-check the manuscript again.

Thanks for the comment. English was corrected.

Round 2

Reviewer 1 Report

I have no further commons

Author Response

Thanks for the comments that really helped to improved the manuscript.

Reviewer 3 Report

I am satisfied with the answers provided by the authors in general. I have just one last remark. 

In comment 6, it is good that authors have provided F0 values for blank measurements. I would only like to confirm if there was no photocurrent change in blank samples when ATP was introduced.

Author Response

Thanks for the comment. In line 300 and 347 of the manuscript we clarify that not change of photocurrent was observed when only ATP was incubated in the channels functionalized with PHEMA-aptamer fragments or PHEMA-aptamer layers.
